# Optimization of the Acid Cleavage of Proanthocyanidins and Other Polyphenols Extracted from Plant Matrices

**DOI:** 10.3390/molecules28010066

**Published:** 2022-12-21

**Authors:** Jesus N. S. Souza, Tatiana Tolosa, Bruno Teixeira, Fábio Moura, Evaldo Silva, Hervé Rogez

**Affiliations:** 1Center for the Valorization of Amazonian Bioactive Compounds (CVACBA), Federal University of Pará (UFPA), Belém 66075-750, PA, Brazil; 2Institute of Coastal Studies, Federal University of Pará (UFPA), Bragança 68600-000, PA, Brazil

**Keywords:** astringency, *Byrsonima crassifolia*, *Euterpe oleracea*, *Inga edulis*, Amazonian plant

## Abstract

The chemical mechanism of the acid cleavage of proanthocyanidins (PAs) has been known for decades but has yet to be optimized. Therefore, we optimized this process in *Byrsonima crassifolia*, *Euterpe oleracea* and *Inga edulis* extracts using the response surface methodology and assessed the effect of hydrochloric acid concentration (0.3–3.7 N), time (39–291 min), and temperature (56–98 °C) on the following response variables: PAs reduction, astringency reduction, antioxidant capacity/total polyphenols (TEAC/TP) ratio, and cyanidin content. The response variables were maximized when cleavage was performed with 3 N HCl at 88 °C for 165 min. Under these conditions, the mean PAs value and astringency in the three extracts decreased by 91% and 75%, respectively, the TEAC/TP ratio remained unchanged after treatment (*p* > 0.05), and the increase in cyanidin confirmed the occurrence of cleavage. Thus, the results suggest that acid cleavage efficiently minimizes undesirable technological PAs characteristics, expanding the industrial applications.

## 1. Introduction

Tannins, which are polymers of phenolic compounds widely distributed in plants and molecular weights ranging between 500 and 30,000 Daltons, play an important role in plant growth by providing protection against predators. These compounds are classified in accordance with different chemical structures of the monomeric units into condensed, hydrolysable, or complex tannins. Also known as proanthocyanidins (PAs), condensed tannins are oligomers, polymers, or high polymers formed by oxidative condensation of flavan-3-ol units. The acid cleavage of PAs yields anthocyanidins with specific names [1,2,3].

PAs have several industrial (e.g., green chemicals, tanning, dyeing, and medicines) and biological applications (health-promoting properties due to scavenging of free radicals and inhibition of lipid peroxidation) [1,4]; however, PAs contained in plant-derived products have an influence on the sensory perception of bitterness/astringency and nutritional values that limits its industrial use. More specifically, these compounds can bind to salivary proteins, carbohydrates, metal ions, and digestive enzymes, which reduces the nutritional value and digestibility of foods [5,6]. Therefore, industrial processes that reduce the degree of polymerization (DP) are needed to increase PAs digestibility and bioavailability. In fact, in vivo studies have shown that PAs with DP lower than four can be absorbed in the human digestive tract. Although the majority of PAs is metabolized by microbial fermentation, its correlation with potential health effects is still not clear [7].

Due to the large biodiversity and equatorial location of Amazonia, many plants produce high content of antioxidants for protection against UVA, UVB, and microbial attack. Among them, phenolic compounds are largely metabolized [8,9,10]. In order to avoid deforestation, multi-strata agroforestry systems have been used by small Amazonian farmers to grow various agricultural and forestry crops, with emphasis on permanent crops such as açai (*Euterpe oleracea*), inga (*Inga edulis*), and muruci (*Byrsonima crassifolia*), with high economic and nutritional importance for the local population. Due to the high content of bioactive compounds, these matrices have been highlighted in several studies involving the prospection of phenolic compounds [8,11,12]. These compounds are highly sought after by the cosmetic (photochemical protectors), pharmaceutical (to prevent several chronic degenerative diseases), and food industries (antioxidants and natural coloring) [13].

Industrial processes of extracting Pas from plants are well reported [14]. Pas are routinely extracted by using water, organic solvents, and subsequent purification; however, other processes also use yeast fermentation [15] or alkaline hydrolysis [16]. In order to increase health promotion, the hydrolysis of PAs can be performed by thermal decomposition (50–100 °C) under acidic conditions (hydrochloric, sulfuric or 0.1–2 N nitric acid) with low alcohol content (propanol, butanol, pentanol, and isopentanol) [17]. Nonetheless, an optimized process to obtain PAs oligomers from plant extracts is still lacking and becomes of great industrial interest [18].

The aims of this study were to (i) optimize the acid cleavage of PAs in the hydro-ethanolic plant extracts of *B. crassifolia* leaves, *I. edulis leaves*, and *E. oleracea* fruit by using the response surface methodology (RSM); (ii) quantify the cyanidin content, produced after cleavage; (iii) evaluate the effect of cleavage on astringency reduction; and (iv) evaluate the antioxidant capacity/total polyphenols (TEAC/TP) ratio.

## 2. Results and Discussion

### 2.1. Assessment of the Effect of Alcohol on Cleavage

The first experiment used *I. edulis* extract to identify the optimal alcohol content and type (methanol or ethanol) for PAs cleavage, and the results are shown in Table 1.

For PAs reduction, the addition of at least 60% alcohol (methanol or ethanol) favors the cleavage reaction, which may be attributable to the following: oligomeric PAs are easily solubilized in solutions consisting of water and organic solvents [19], and the solubilization of these molecules favors the electrophilic attack of the H+ ion, the first cleavage reaction step. The reduction in PAs did not significantly differ between the types of alcohol tested.

The largest total polyphenol content was observed in the reaction medium containing 60% ethanol. This behavior is due to the formation of nonpolar compounds during the acid treatment that are more soluble in ethanol than in methanol [20]. Notably, the most nonpolar compounds (monomeric aglycone flavonoids) result from both PAs cleavage and glycosylated flavonoid hydrolysis. The dielectric constant (DC), which describes the ability of the solvent to insulate opposite charges, is a measure of solvent polarity [21]. Based on the DC values, ethanol (DC = 24) is more nonpolar than methanol (DC = 33). Thus, a reaction medium containing ethanol favors the solubilization of nonpolar compounds produced by PAs cleavage and the hydrolysis of glycosylated flavonoids. Therefore, we used ethanol at a concentration of 60% to optimize the cleavage process.

### 2.2. Optimization of Acid Cleavage by RSM

Table 2 shows the PAs reduction, astringency reduction, TEAC/TP ratio, and cyanidin content for various experimental conditions.

Specifically, the experimental conditions elicited a wide range of PAs reductions, with values from 30 to 95% for the *B. crassifolia* extract, 36% to 97% for the *E. oleracea* extract, and 2 to 91% for *I. edulis* extract.

Generally, the degree of polymerization and molecular weight of the phenolic compound directly correlate with astringency [5]. Thus, evaluating the effect of PAs cleavage on the extract’s astringency is important. The results presented in Table 2 show that the acid cleavage treatments reduced the astringency of extracts. Specifically, the astringency of *B. crassifolia* extract was minimized (91.8% of reduction) when the extract was treated with 3 N HCl at 90 °C for 240 min (assay 8). The compounds responsible for this plant’s astringency are sensitive to the treatments employed because the minimum reduction value (~82%) was achieved using milder treatments. The reduction in the astringency of *E. oleracea* extract varied from 0% (assay 9) to 91% (assay 10), whereas that of *I. edulis* extract ranged between 2.85% (assay 9) and 85% (assay 8).

The impact of acid treatment on the antioxidant capacity of the extracts was evaluated based on the ratio of the antioxidant capacity to the total polyphenol content. The TEAC/TP values of *B. crassifolia* or *E. oleracea* extracts did not significantly differ by treatment (*p* > 0.05), which is from a technological point of view because it indicates that the severity of the experimental conditions did not affect the antioxidant activity of these extracts.

The presence of anthocyanidin after the acid treatment demonstrates the occurrence of PAs cleavage [22]. In the *E. oleracea* extract, cyanidin originated from PAs cleavage and the hydrolysis of its two glycosylated cyanidins (3-rutinoside and 3-glucoside) [23].

#### 2.2.1. Multiple Linear Regression Analysis of the Experimental Data

The fit of the experimental data to the second-order polynomial model was assessed using an analysis of variance (ANOVA) and the lack-of-fit test (Table 3).

The ANOVA of the model was significant (*p* < 0.05) for all variables except TEAC/TP for the *E. oleracea* extract, suggesting that at least one of the evaluated variables influences the PAs reduction, astringency reduction, TEAC/TP, and cyanidin content of the extracts. The behavior of the *E. oleracea* extract indicates that the treatments used to cleave PAs did not significantly change the TEAC/TP ratio. Thus, the effect of polymeric compounds on the TEAC/TP was attenuated by the monomeric compounds produced during the acid cleavage of PAs and hydrolysis of glycosylated flavonoids. Specifically, aglycone flavonoids have been shown to possess higher antioxidant capacity than their corresponding glycosylated forms [24].

The lack-of-fit test was not significant (*p* > 0.05) for any of the response variables, indicating that the model fitted the experimental data well; good coefficients of determination (R^2^) were obtained for PAs reduction (>0.94), astringency reduction (>0.77), and cyanidin content (>0.93). The TEAC/TP ratio exhibited low R^2^ values for the *B. crassifolia* (0.63) and *I. edulis* (0.49) extracts, indicating that the model inadequately explained the variability of the TEAC/TP response.

#### 2.2.2. Analysis of the Coefficients of Determination and Response Surface

The coefficients of determination for the second-order polynomial model describing the PAs reduction, astringency reduction, and cyanidin content of plant extracts are shown in Table 4.

According to the model proposed, the reduction in PAs in the three extracts depends on the linear terms of the variables HCl concentration (*p* < 0.001), temperature (*p* < 0.001), and time (*p* < 0.05), the quadratic terms of HCl concentration and temperature (*p* < 0.01 for *B. crassifolia* and *I. edulis*, and *p* < 0.05 for *E. oleracea*), and the interaction between the HCl concentration and temperature variables (*p* < 0.05). The quadratic effect of time also affects the *I. edulis* extract (*p* < 0.01).

The intercept value represents the response variable at the design central point (2 N HCl, 77.5 °C for 165 min.) and was approximately 78% for PAs reduction in all three extracts. The positive effect of the linear terms indicates that increasing the level of any variable further reduces the content of PAs. The role of HCl concentration is explained by its action as a reaction catalyst, i.e., it protonates the cleavage site of the interflavanoid bond [25]. In the proposed model, increasing the HCl concentration from level 0 to level 1 (i.e., from 2 N to 3 N) further reduces the PAs level by an additional 15.8%, 13.8%, and 19.4% in the *B. crassifolia*, *E. oleracea*, and *I. edulis* extracts, respectively. The linear positive effect of temperature is similar to the acid concentration and can be explained by the Arrhenius Law [26]. Specifically, the PAs level is reduced a further 13% in *B. crassifolia* and *E. oleracea* extracts and a further 23% in *I. edulis* extract when the temperature is increased by 12.5 °C.

The negative quadratic effects of time on the *I. edulis* extract and HCl concentration and temperature on the *B. crassifolia*, *E. oleracea*, and *I. edulis* extracts indicate that any increases in their values beyond the peak value inhibit PAs reduction. Moreover, the negative effect of the HCl concentration x temperature interaction indicates that the simultaneous increase in the level of the two input variables decreases PAs reduction. Thus, individual temperature variations at constant minimum levels of acid concentration significantly increase the PAs reduction (Figure 1), showing a direct effect on acid cleavage kinetics. 

According to Table 4, the linear terms of HCl concentration and temperature positively affect astringency reduction in all three extracts. Moreover, the linear effect of time was significant in the *B. crassifolia* and *I. edulis* extracts. The quadratic effect of temperature was significant for all three extracts. The quadratic effect of HCl concentration and the interaction between HCl concentration and temperature was significant only for the *B. crassifolia* and *I. edulis* extracts.

The astringency reduction differed by extract, as shown by the intercept values. Specifically, the same experimental conditions (HCl 2 N, 77.5 °C for 165 min.) result in different degrees of astringency reduction: 84%, 58%, and 48% for *B. crassifolia*, *E. oleracea*, and *I. edulis*, respectively. The effect of the treatment becomes evident when the results are presented as response surfaces (Figure 1), which show that the *B. crassifolia* extract behaves differently from the *E. oleracea* and *I. edulis* extracts. Specifically, the astringency reduction of the *B. crassifolia* extract, although significant, was not greatly influenced by HCl concentration or temperature. This finding is confirmed by the small coefficients of determination of these variables (Table 4) and the small variation in astringency reduction (from 81.7 to 91.8%) for different assay conditions.

The uniqueness of extract behavior can be attributed to differences in the phenolic profile of the extracts (three different species). Specifically, differences in the spatial configuration of flavanols significantly affect the intensity of astringency, which is greater for epicatechin than in catechin. Regarding PAs, the type of monomer and the location of the bond between units affects astringency; for instance, the B6 dimer (catechin-4,6-catechin) is more astringent than the B3 (catechin-4,8-catechin) and B4 (catechin-4,8-epicatechin) dimers. Another factor contributing to the increased astringency of phenolic compounds is the esterification with gallic acid. Generally, any chemical structure that promotes an increase in the number of bonds between the hydroxyl groups of the polyphenol and the carbonyls of salivary proteins enhances the perception of astringency [27,28].

Based on the intercept values shown in Table 4, the cyanidin contents of the three extracts differed. Specifically, the same experimental condition (2 N HCl, 77.5 °C for 165 min.) resulted in cyanidin contents of 14.2 mg/g DE, 95.4 mg/g DE, and 5.37 mg/g DE for *B. crassifolia*, *E. oleracea*, and *I. edulis*, respectively. The high cyanidin content in *E. oleracea* is due to the formation of cyanidin from both the hydrolysis of glycosylated anthocyanins and the cleavage of PAs. The difference between the cyanidin contents of the *B. crassifolia* and *I. edulis* extracts is due to significant differences in the degree of polymerization between these compounds (*p* > 0.05). For instance, a cleaved trimer produces two anthocyanidin molecules and one flavanol molecule, whereas this ratio is 1:1 in a dimer, i.e., one anthocyanidin and one flavanol molecule [29,30].

The coefficients of determination show that the linear effect was significant for the HCl concentration (*B. crassifolia* and *E. oleracea*), temperature (*B. crassifolia*, *E. oleracea*, and *I. edulis*), and time (*I. edulis*) variables. This behavior indicates that an increase in any of these variables favors PAs reduction via cleavage and the subsequent formation of cyanidin. The negative quadratic effects of HCl concentration for the three extracts, temperature for the *E. oleracea* extract, and time for the *I. edulis* extract indicate that successive increases in the level of these variables starting at the surface curvature point promote cyanidin degradation. This behavior is shown in Figure 1.

#### 2.2.3. Determination of Optimal Conditions and Model Validation

To assess the suitability of the model for predicting the response variables, an optimum condition was determined based on the maximum D of PAs reduction, astringency reduction, and cyanidin content. This choice was driven by experimental conditions under which the PAs and astringency reductions exceeded 90% and 70%, respectively. The optimization of the response variables predicts the use of 3 N HCl under heating at 88 °C for 165 min. for the three plant extracts. Table 5 shows the predicted and experimental values for the response variables.

The experimental values for PAs reduction, astringency reduction, and cyanidin content for the three extracts are within the confidence interval of their respective predicted values. This finding shows that the proposed model adequately explains the influence of the HCl concentration, temperature, and time on the behavior of the response variables.

## 3. Materials and Methods

### 3.1. Crude Extracts from Plant Matrices

The plant sources used in this study were kindly provided by Amazon Dreams^®^ (Belém, Brazil). The crude extract from dried *B. crassifolia* and *I. edulis* leaves or *E. oleracea* fruits was extracted with ethanolic solution at 50 °C [31,32]. The crude extract was concentrated, and the phenolic compounds were adsorbed to and desorbed from macroporous resin [33,34]. The resultant extract was concentrated again until all ethanol was eliminated. At the end of the process, the *B. crassifolia*, *I. edulis*, and *E. oleracea* dry matter contents were 15.7%, 5.8%, and 8.6%, respectively.

### 3.2. Determination of Total Polyphenol Content

The TP content was determined using the Folin–Ciocalteu colorimetric method [35]. Quantification was performed in triplicate, and the calibration curve was obtained using gallic acid as a standard. The results are expressed as milligrams of gallic acid equivalents per gram of dry extract (mg GAE/g DE).

### 3.3. Determination of Proanthocyanidin Content

The PAs content was determined using the butanol-HCl method [36]. Quantification was performed in triplicate, and based on a calibration curve obtained with cyanidin standard. The content of PAs was determined as milligrams of cyanidin equivalents per gram of dry extract (mg CyE/g DE), and expressed as percent PAs reduction.

### 3.4. Evaluation of Antioxidant Capacity

The TEAC (Trolox Equivalent Antioxidant Capacity) method was used to evaluate the antioxidant activity of the extracts [37]. Assays were performed in microplates in triplicate, and the results are expressed in micromoles of Trolox equivalents per gram of dry extract (µmol TE/g DE).

### 3.5. Determination of Astringency

Astringency was determined using the turbidimetric method to assess sensory changes in the extracts due to the actual cleavage of PAs [38]. The equipment was calibrated with a formazin standard before the readings. The results were obtained as nephelometric turbidity units per gram of dry extract (NTU/g DE), and are expressed as percent astringency reduction.

### 3.6. Chromatographic Analysis

The high-performance liquid chromatography (HPLC) system used in this study was a Shimadzu LC10AVP series instrument (Tokyo, Japan) consisting of the following: a DGU 14A degasser, an LC10AT quaternary pump, an SIL10AF auto-injector, a CTO10AS oven, an SPDM20A diode array detector, and CLASS VP chromatography data station software. Analyses were performed using a Gemini C18 column (Phenomenex, Torrance, CA, USA) (150 × 4.6 mm column, with 3 μm average particle diameter) coupled to a 3 × 4 mm pre-column (Phenomenex, Torrance, CA, USA) maintained at 30 °C.

The HPLC analysis method was based on Souza et al. [38]. The mobile phase consisted of ultrapure water acidified with 1% formic acid (solvent A) and acetonitrile acidified with 1% formic acid (solvent B). The injected sample volume was 20 μL, with a mobile phase flow of 1 mL. min^−1^. Elution was performed according to the following gradient: 73.5% B for 26 min, 35.7% B for 6 min, and 7% B for 3 min. Cyanidin was quantified using a calibration curve obtained with a cyanidin standard (Extrasynthèse, Genay, France) measured at 515 nm.

### 3.7. Acid Cleavage of Proanthocyanidins

#### 3.7.1. Assessment of the Effect of Alcohol on Cleavage

To determine the type and the amount of alcohol, cleavage assays were performed using *I. edulis* leaf extract containing 0%, 30%, and 60% of each alcohol (ethanol and methanol) in the presence of 0.1 N HCl at 90 °C for 240 min. The decrease in PAs in relation to the content in crude extract (100%) and after cleavage (%), and the total polyphenol content (mg GAE/g DE) were evaluated.

#### 3.7.2. Optimization of Acid Cleavage: Experimental Design

The acid cleavage of PAs on *B. crassifolia*, *I. edulis*, and *E. oleracea* was optimized using the RSM. A rotational central composite design (RCCD) was used with four replicates at the center point to assess the effect of the HCl concentration, reaction time, and reaction temperature on the following response variables: PAs reduction (%), astringency reduction (%), TEAC/TP ratio (µmol TE/g GAE), and cyanidin content (mg/g DE).

The assays were performed by diluting the crude extracts according to the design (ethanol:water (6:4), acidified with HCl) with heating in a water bath. All assays were conducted in test tubes with screw lids and a Teflon septum, and all samples were immediately cooled in an ice bath in the end of experiment.

### 3.8. Statistical Analysis

The results obtained were analyzed using analysis of variance (ANOVA) and Tukey’s test to determine the difference between means, at a significance level of 95% (*p* < 0.05), using the STATISTIC 7.0 software (StatSoft, Tulsa, OK, USA).

A multiple linear regression analysis of the data was performed in STATISTICA 7.0. The data were mathematically modeled with a second-order polynomial model (Equation (1)).
(1)$$Y=\beta_{0}+\sum_{i=1}^{k}\beta_{i}X_{i}+\sum_{i=1}^{k}\beta_{ii}X_{i}^{2}+\sum_{\begin{array}{l} i=1\\ i<j \end{array}}^{k-1}\sum_{j=2}^{k}\beta_{ij}X_{i}X_{j}$$
where X1, X2... Xk are independent variables that affect the response variable Y; β_0_, βi (i = 1, 2,..., k), β_ii_ (i = 1, 2,..., k), and β_ij_ (i = 1, 2,..., k; j = 1, 2,..., k) are the coefficients for the intercept, linear, quadratic, and interactions parameters, respectively; k is the number of variables.

In the statistical treatment of the experimental data, the input variables (HCl concentration, time, and temperature) were maintained at their normalized values.

### 3.9. Determination of Optimal Conditions and Model Validation for Acid Cleavage of Polyphenols

The optimal cleavage conditions were selected according to the desirability values (D). The value of D varies directly and proportionately with the response variable and ranges from 0 (undesirable response) to 1 (desirable response) [39,40]. Thus, the optimal HCl concentration, reaction temperature, and reaction time yield a value of D that minimizes the PAs content, minimizes the astringency, optimizes the TEAC/TP ratio, and optimizes the cyanidin content, i.e., a D equal to or close to 1.

To validate the chosen optimal condition, five replicate assays were carried out. The extract used in the validation was also used to carry out the experimental design assays. The response variable values obtained experimentally were compared with those predicted by the model to verify its suitability for expressing the behavior of the experimental results.

## 4. Conclusions

The use of the RSM allowed to optimize the cleavage of PAs in the hydro-ethanolic extracts of three plant matrices (*B. crassifolia*, *I. edulis*, and *E. oleracea*). The maximization of the response variables was verified using a second-order polynomial model: PAs reduction, astringency reduction, and cyanidin content were optimized when cleavage was performed with 3 N HCl with heating at 88 °C for 165 min. Notably, the optimal condition was the same for all three plant extracts, suggesting that the cleavage of interflavonoid bonds is independent of the plant matrix (*B. crassifolia*, *E. oleracea*, or *I. edulis*), the nature of the extract (leaf or fruit), and the profile of phenolic compounds.

Under the conditions proposed by the model, the acid treatment resulted in the following: (*i*) the depolymerization of PAs and the hydrolysis of glycosylated flavonoids, which decreased the polarity of the extracts; (*ii*) the reduction of astringency, one of the factors that limits the use of plant extracts in food; (*iii*) anthocyanidin formation during acid cleavage; and (*iv*) the conservation of the antioxidant capacity of the extracts. These changes extend the potential applications of these extracts, allowing their use in the food industry as a natural antioxidant, including in the oil and high-fat foods area, due to the formation of nonpolar compounds during cleavage.

## Figures and Tables

**Figure 1 molecules-28-00066-f001:**
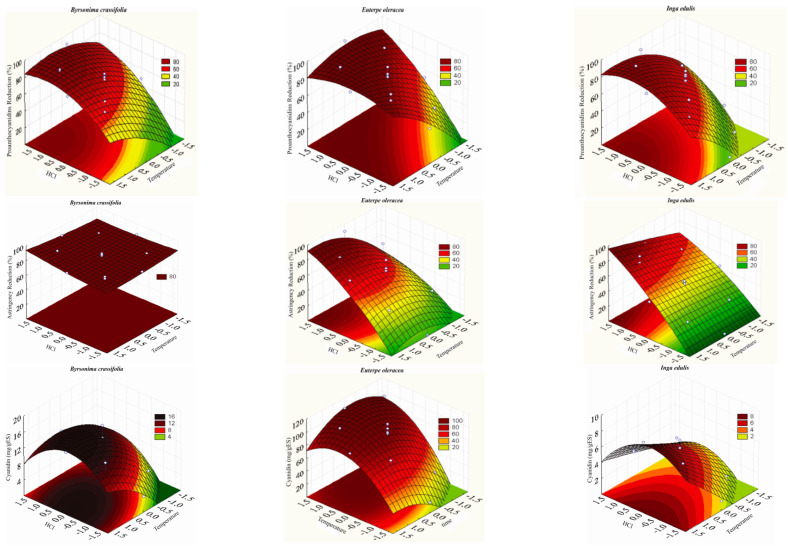
Response surface and contour line for proanthocyanidin reduction, astringency reduction, and cyanidin content in *B. crassifolia*, *E. oleracea*, and *I. edulis* extracts.

**Table 1 molecules-28-00066-t001:** Effect of the type and proportion of alcohol on the acid cleavage of proanthocyanidins in *I. edulis* leaf extract. The different superscript letters within each column refer to statistically significant differences (*p* < 0.05) as resulting from ANOVA followed by Tukey testing.

Alcohol (Type, %)	Reduction in Proanthocyanidins (%)	Total Polyphenols (mg GAE/g DE ^a^)
-	26.0 ^A,B^	241 ^A^
Methanol 30%	25.2 ^A^	366 ^B^
Methanol 60%	32.0 ^B,C^	367 ^B^
Ethanol 30%	27.2 ^A,B^	405 ^B,C^
Ethanol 60%	36.2 ^C^	436 ^C^

^a^ Milligrams of gallic acid equivalents per gram of dry extract.

**Table 2 molecules-28-00066-t002:** Experimental conditions and results of the response variables obtained after the completion of the experimental design.

Assay	Experimental Conditions ^a^			Reduction in Proanthocyanidins (%)			Reduction of Astringency (%)			TEAC/TP (μmol TE/g GAE ^b^)			Cyanidin (mg g^−1^ Dry Extract)		
	HCl (N)	T (°C)	Time (min)	BC ^c^	EO ^d^	IE ^e^	BC	EO	IE	BC	EO	IE	BC	EO	IE
1	1.00 (−1)	65.00 (−1)	90.00 (−1)	32.04	36.18	2.04	81.70	33.68	10.50	2415	3207	3303	1.75	5.80	1.17
2	1.00 (−1)	65.00 (−1)	240.00 (+1)	33.24	39.89	9.47	84.31	33.96	22.95	2767	3778	3088	5.15	13.44	1.74
3	1.00 (−1)	90.00 (+1)	90.00 (−1)	63.78	80.25	58.95	81.51	22.04	25.45	1617	3759	3234	13.58	68.37	6.52
4	1.00 (−1)	90.00 (+1)	240.00 (+1)	76.28	87.73	79.34	84.02	39.78	25.16	1679	2923	2571	11.70	71.86	5.65
5	3.00 (1)	65.00 (−1)	90.00 (−1)	65.21	73.50	43.95	82.79	68.72	59.54	1300	3379	3436	9.65	31.80	3.84
6	3.00 (1)	65.00 (−1)	240.00 (+1)	69.72	82.66	60.13	84.17	54.13	62.29	1143	4163	2681	10.32	87.84	2.47
7	3.00 (1)	90.00 (+1)	90.00 (−1)	87.75	89.44	84.34	85.37	82.14	76.01	1461	4212	3643	12.93	115.98	5.42
8	3.00 (1)	90.00 (+1)	240.00 (+1)	89.55	91.85	91.38	91.78	60.67	84.64	1444	4143	4462	13.46	104.33	5.24
9	0.32 (−1.68)	77.50 (0)	165.00 (0)	29.85	40.70	5.33	84.75	0.14	2.85	2821	3602	3078	3.73	16.90	1.11
10	3.68 (+1.68)	77.50 (0)	165.00 (0)	94.98	97.23	85.26	87.88	91.25	77.32	2866	4901	2720	10.92	108.50	2.84
11	2.00 (0)	56.48 (−1.68)	165.00 (0)	48.26	50.73	16.51	85.59	3.24	43.59	2351	4711	2607	5.09	15.34	1.15
12	2.00 (0)	98.52 (+1.68)	165.00 (0)	80.08	87.11	83.75	86.13	76.07	49.02	2344	6076	2349	15.53	104.68	8.91
13	2.00 (0)	77.50 (0)	38.87 (−1.68)	62.11	70.64	38.22	83.89	39.31	44.05	2543	5480	2644	10.11	49.29	3.96
14	2.00 (0)	77.50 (0)	291.13 (+1.68)	79.81	89.97	40.66	86.43	76.89	49.35	2946	5536	2508	11.81	95.03	4.38
15	2.00 (0)	77.50 (0)	165.00 (0)	80.54	81.79	77.50	83.54	50.51	46.93	3412	3611	2600	13.54	94.08	4.81
16	2.00 (0)	77.50 (0)	165.00 (0)	77.22	78.94	79.01	83.35	62.18	50.13	2584	5087	2460	13.88	95.86	5.50
17	2.00 (0)	77.50 (0)	165.00 (0)	76.92	75.66	72.04	83.33	65.57	49.87	3412	4689	2604	16.95	101.72	4.74
18	2.00 (0)	77.50 (0)	165.00 (0)	73.82	78.38	83.36	84.70	52.03	45.97	3801	5172	2571	12.68	89.89	6.50

^a^ Values in parentheses are normalized. ^b^ Micromoles of Trolox equivalents per gram of gallic acid equivalents. ^c^
*B. crassifolia*. ^d^
*E. oleracea*. ^e^
*I edulis*.

**Table 3 molecules-28-00066-t003:** Analysis of variance and coefficients of determination obtained from experimental data for the response variables.

Proanthocyanidins Reduction (%)							
		*B. crassifolia*		*E. oleracea*		*I. edulis*	
Source	DF ^a^	SS ^b^	F ^c^	SS	F	SS	F
Model	9	6420	873.3 *	6008.5	949 *	16,238.6	744 *
Lack-of-fit	5	258.8	6.9 ^NS^	177.6	4.1 ^NS^	799.6	7.3 ^NS^
Pure error	3	22.6	19.0			65.48	
R^2 d^		0.9580	0.9682			0.9464	
**Astringency Reduction (%)**							
		** *B. crassifolia* **		** *E. oleracea* **		** *I. edulis* **	
Source	DF	SS	F	SS	F	SS	F
Model	9	78.9	185.2 *	8729	158 *	8204.3	1881 *
Lack-of-fit	5	17.5	6.3 ^NS^	2332.6	8.4 ^NS^	178.3	8.2 ^NS^
Pure error	3	1.3		165.8		4.4	
R^2^		0.8046		0.7770		0.9772	
**TEAC/TP Ratio (μmol TE/g GAE ^e^)**							
		** *B. crassifolia* **		** *E. oleracea* **		** *I. edulis* **	
Source	DF	SS	F	SS	F	SS	F
Model	9	861,002	33 *	5,360,140	10 ^NS^	3,033,826	37 *
Lack-of-fit	5	3,149,084	2.4 ^NS^	6,550,309	2.5 ^NS^	2,475,097	6.0 ^NS^
Pure error	3	788,657		1,545,272		248,910	
R^2^		0.6288		0.4045		0.4917	
**Cyanidin content (mg/g Dry extract)**							
		** *B. crassifolia* **		** *E. oleracea* **		** *I. edulis* **	
Source	DF	SS	F	SS	F	SS	F
Model	9	308.7	89 *	25,038.8	1041.9 *	74.6	111.3 *
Lack-of-fit	5	4.9	0.3 ^NS^	730.0	6.1 ^NS^	3.2	1 ^NS^
Pure error	3	10.4		72.2		2	
R^2^		0.9490		0.9674		0.9341	

^a^ Degrees of freedom. ^b^ Sum of squares. ^c^ Fisher’s exact test. ^d^ Regression coefficient. ^e^ Micromoles of Trolox equivalents per gram of gallic acid equivalents. * Significant at *p* < 0.05. ^NS^ Not significant.

**Table 4 molecules-28-00066-t004:** Regression coefficients for the parameters of the second-order polynomial model for the response variables.

	*B. crassifolia*		*E. oleracea*		*I. edulis*	
Model parameter	RC ^a^	SE ^b^	RC	SE	RC	SE
Intersect	77.14 ***	1.37	78.7 ***	1.26	77.6 ***	2.33
HCl	15.85 ***	0.74	13.80 ***	0.68	19.36 ***	1.26
HCl^2^	−5.3 **	0.77	−3.40 *	0.71	−9.23 **	1.31
T	12.50 ***	0.74	13.05 ***	0.68	22.81 ***	1.26
T^2^	−4.68 **	0.77	−3.4 *	0.71	−7.5 *	1.31
Time	3.65 *	0.74	4.04 **	0.68	4.04 *	1.26
Time^2^	−2.28 ^NS^	0.77	0.64 ^NS^	0.71	−11.30 **	1.31
HCl × T	−4.05 *	0.97	−8.35 **	0.89	−6.89 *	1.65
HCl × Time	−0.92 ^NS^	0.97	0 ^NS^	0.89	−0.57 ^NS^	1.65
T × Time	1.07 ^NS^	0.97	−0.37 ^NS^	0.89	0.48 ^NS^	1.65
	*B. crassifolia*		*E. oleracea*		*I. edulis*	
Model Parameter	RC	SE	RC	SE	RC	SE
Intersect	84 ***	0.33	58.17 ***	3.71	48.1 ***	1.04
HCl	1.31 **	0.18	21.19 **	2.01	23.70 ***	0.56
HCl^2^	0.60 *	0.18	−3.86 ^NS^	2.09	−2.51 *	0.59
T	0.78 *	0.18	10.0 *	2.01	4.77 **	0.56
T^2^	0.40	0.18	−6.0 ^NS^	2.09	−0.31 ^NS^	0.59
Time	1.26 **	0.18	3.31 ^NS^	2.01	2.38 *	0.56
Time^2^	0.2 ^NS^	0.18	0.65 ^NS^	2.09	−0.17 ^NS^	0.59
HCl × T	1.33 **	0.23	3.22 ^NS^	2.63	2.71 *	0.74
HCl × Time	0.33 ^NS^	0.23	−6.76 ^NS^	2.63	−0.01 ^NS^	0.74
T × Time	0.62 ^NS^	0.23	1.32 ^NS^	2.63	−0.86 ^NS^	0.74
	*B. crassifolia*		*E. oleracea*		*I. edulis*	
Model parameter	RC	SE	RC	SE	RC	SE
Intersect	14.24 ***	0.93	95.41 ***	2.45	5.37 ***	0.41
HCl	1.92 *	0.50	24.44 ***	1.33	0.35 ^NS^	0.22
HCl^2^	−2.33 *	0.52	−11.68 **	1.38	−1.12 *	0.23
T	3.10 **	0.50	27.27 ***	1.33	1.95 **	0.22
T^2^	−1.27 ^NS^	0.52	−12.64 **	1.38	−0.04 ^NS^	0.23
Time	0.41 ^NS^	0.50	9.66 **	1.33	−0.08 ^NS^	0.22
Time^2^	−1 ^NS^	0.52	−8.34 **	1.38	−0.35 ^NS^	0.23
HCl × T	−1.50 ^NS^	0.66	−2.60 ^NS^	1.73	−0.61 ^NS^	0.29
HCl × Time	0 ^NS^	0.66	4.22 ^NS^	1.73	−0.16 ^NS^	0.29
T × Time	0 ^NS^	0.66	−9 *	1.73	−0.03 ^NS^	0.29

^a^ Regression coefficient. ^b^ Standard error. *, **, and *** Significant at *p* < 0.05, 0.01, and 0.001, respectively. ^NS^ Not significant.

**Table 5 molecules-28-00066-t005:** Predicted and experimental results for the response surface methodology under the optimal conditions.

Response Variable	*B. crassifolia*		*E. oleracea*		*I. edulis*	
	Vpred ^a^	Vexp ^b^	Vpred	Vexp	Vpred	Vexp
Proanthocyanidin Reduction (%)	91.5 ± 11.2	90.2 ± 2.1	90.7 ± 9.3	94.4 ± 1.0	95.7 ± 19.5	91.78 ± 3.2
Astringency Reduction (%)	87.8 ± 3	79.3 ± 6.7	81.9 ± 33.2	77.2 ± 5.3	75.4 ± 9.2	77.8 ± 7.8
Cyanidin content (mg/g DE ^c^)	14.3 ± 2.6	11.1 ± 1.4	120 ± 18.8	110 ± 6.3	5.7 ± 1.5	8.4 ± 1.8

^a^ Value predicted by the model ± 95% confidence interval. ^b^ Experimental value ± standard deviation. ^c^ Dry extract.

## Data Availability

All data are available in the article.

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
