# Peer review of "Optimization of the Acid Cleavage of Proanthocyanidins and Other Polyphenols Extracted from Plant Matrices"

_molecules, 2022, doi:10.3390/molecules28010066_

Round 1

Reviewer 1 Report

The paper reports optimization of the acid cleavage of polyphenols. The results have some meaning for expanding the industrial applications. However, some parts of this paper need to be revised.

1) Title:according to the method of sample preparation, the sample contained not only proanthocyanidins, but also other phenolic compounds. Therefore, the title should be revised.

2) Why did the authors choose the B. crassifolia, I. edulis, and E. oleracea?

3) The authors set antioxidant capacity/total polyphenols (TEAC/TP) ratio as one response variable. It is not reasonable. Generally, the higher content of TP, the stronger antioxidant capacity. Therefore, the ration will not change much after the treatment. the results should be reanalyzed.

4) The “Statistical analysis” section should be added.

Author Response

We thank the reviewer one for the positive comments on this paper, Please see the attachment with responses for your sugestions. 

Reviewer 2 Report

Manuscript title: Optimisation of the acid cleavage of proanthocyanidins extracted from plant matrices

Manuscript ID: Molecules-2039127

General comments:

The manuscript describes the statistical optimization procedure adopted to figure out the conditions for hydrolyzing proanthocyanidins. The manuscript is well-written and may be of industrial importance. However, the following queries are to be addressed before the manuscript is considered for publication. Also, there are a few grammatical errors in the manuscript.

Specific comments:

[1] The authors have not produced any data on the cleavage products

[2] Table 1: How is the reduction (in percentage) of proanthocyanidins calculated?

[3] Visible changes in the reduction of proanthocyanidins may be due to the higher hydrophobicity of ethanol which results in higher solubility of PA or its hydrolyzed products and may not have a direct impact on the acid cleavage. This is evident from the increased concentration of total polyphenols upon changing the solvent

[4] Is there any evidence that the higher ethanol content has the same effect on the extraction / acid cleavage of the other plant extracts? Table 1 contains only the information on I. edulis.

[5] The type of statistical test depicted in Table 1 can be explicitly mentioned

[6] Any reason for the observed reduction in the astringency of the B. crassifolia extracts? Are there any changes in the digestion or hydrolysis profiles? How is the completion of the hydrolytic reaction confirmed?

[7] The rationale involved in obtaining a different percentage of PAs reduced and the total phenolics from different plant sources should be detailed. Moreover, the improved characteristics of the proanthocyanidins, with respect to hydrolysis and decreased astringency, have to be carefully normalized with respect to the total (and extractable) polyphenols obtained from different sources.

[8] Authors should also try to prove the decrease in astringency using biological models, instead of completely relying on chemical-based protocols.

[9] On page 3, line 121, glycosylated derivatives (rutinoside and glucoside) are mentioned. How has this arrived? The experiment used or the reference from the literature should be included.

[10] Enhancement of the nutritional value of cyanidin (or the hydrolyzed products of proanthocyanidin) should be signified. What is their role in alleviating the anti-nutrient capacity of proanthocyanidin?

[11] Regression analysis (maybe a Pearson correlation or use of multivariate analysis) correlating the effect of the experimental variables used on the observable parameters such as hydrolysis and astringency will provide more meaningful information. This could be done at different levels – for a single plant extract and across three different plant extracts used. This may give lead to the general applicability of this methodology for enhancing the nutritive value of the extracts.

[12] In the optimization, how were the minimum and maximum values of the different variables finalized?

[13] The authors should also comment on the use of high temperatures (90 C) on maintaining the concentration of the solution, which might affect the kinetics of the acid cleavage.

[14] The contour plots indicate that the parameters are not yet completely optimized, especially for proanthocyanidin and astringency reduction.

[15] Line 337: A mixed solvent system is used in the study and it is not a complete aqueous extract.

[16] It is also suggested to include the fold change associated with the observed variables before and after the optimization of the parameters.

Author Response

We thank the reviewer 2 for the positive and kind general comment on the study. We made a grammatical correction in the manuscript. Please see the attachment with responses for your sugestions.   

Round 2

Reviewer 2 Report

The authors have responded appropriately and satisfactorily to the queries raised